# Immune-Related Genes in the Honey Bee Mite *Varroa destructor* (*Acarina, Parasitidae*)

**DOI:** 10.3390/insects16040356

**Published:** 2025-03-28

**Authors:** Alfonso Cacace, Giovanna De Leva, Ilaria Di Lelio, Andrea Becchimanzi

**Affiliations:** 1Department of Agricultural Sciences, University of Naples Federico II, 80126 Naples, Italy; alfonso.cacace@unina.it (A.C.); giovanna.deleva@unina.it (G.D.L.); ilaria.dilelio@unina.it (I.D.L.); 2BAT Center—Interuniversity Center for Studies on Bioinspired Agro-Environmental Technology, University of Naples Federico II, 80126 Naples, Italy

**Keywords:** chelicerates, immune response, BLASTp, immunity evolution, gene silencing

## Abstract

*Varroa destructor* is a tiny parasitic mite that threatens honey bee populations worldwide by spreading deadly viruses. While efforts to control these mites often rely on chemical treatments, these can harm beneficial insects and lead to resistant mite populations. A promising alternative is using RNA-based treatments to disrupt essential mite functions, such as immunity. However, scientists still know little about how *Varroa* defends itself against infections. In this study, we analyzed the genetic makeup of *Varroa*’s immune system and compared it to that of related species, such as ticks. Our results show that, similar to ticks, *Varroa* lacks certain immune system components found in other arthropods. Its repertoire of antimicrobial peptides is more closely related to that of ticks and spiders than to that of insects. These findings help us better understand how the mite fights infections and provide a foundation for developing new, targeted methods to weaken and control *Varroa* populations. By improving our knowledge of mite immunity, this research contributes to safer and more effective ways to protect honey bee colonies, ensuring their survival and the vital pollination services they provide to ecosystems and agriculture.

## 1. Introduction

Honey bees (*Apis mellifera*) are facing a serious trend of increasing colony losses caused by a multifactorial syndrome in which parasites and pathogens play a major role [1,2]. As indicated by most surveys, colony losses are associated with high levels of *Varroa destructor* and pathogenic viruses, which are vectored by the mites [3,4]. This arthropod is an obligate ectoparasite of honey bees, and has a negative impact on the host, both at the colony and individual level [5,6], since it feeds on liquid tissues through a wound made on the integument of the host [7,8]. This trophic activity has important consequences for the honey bee host, such as the reduction in longevity and weight at emergence [9,10], and the transmission of viral pathogens [11]. In addition, *Varroa* can also impact reproductive fitness, homing, orientation abilities, and also the immune response of bees [12,13].

With a nearly global presence [11,14], *V. destructor* can significantly weaken or even cause the collapse of honey bee colonies if left untreated [15]. Currently, most beekeepers rely heavily on chemical controls to manage the mite [16], despite their detrimental effects on non-target organisms, their residual presence in hive products, and the insurgence of acaricide resistance [17]. This highlights the importance of adopting an Integrated Pest Management (IPM) approach for *Varroa destructor*, which should combine different control methods, including genetic, mechanical, biological, and biotechnical tools, to reduce the impact of treatments with synthetic pesticides [18].

Additional tools for *Varroa* control include “soft” acaricides, such as oxalic acid and thymol, as well as biopesticides like dsRNA [19]. dsRNA works by inducing the degradation of RNA sequences with similar structures, resulting in targeted gene silencing. This process interferes with the synthesis of the encoded protein, enabling selective targeting of specific organismal functions [20].

dsRNA-based control strategies can be classified as either direct killing, targeting vital genes essential for survival, or indirect killing, where lethality occurs only when combined with another treatment. For instance, targeting immune genes with dsRNA combined with a sublethal dose of an entomopathogen-based biopesticide has been proposed as an indirect killing strategy [21].

RNA interference (RNAi) via dsRNA treatments is effective against a wide range of mite species. Since the first application of RNAi through injection in the spider mite *Tetranychus urticae*, silencing the Distal-less gene [22], an increasing number of herbivore and also predator mite species have been shown to be susceptible to RNAi such as *Dermatophagoides pteronyssinus*, *Metaseiulus occidentalis*, *Panonychus citri*, *Phytoseiulus persimilis*, *Sarcoptes scabiei*, *Tetranychus cinnabarinus*, *Tetranychus urticae*, and *Varroa destructor*. The genome sequencing of *T. urticae* allowed research on mites to enter the genome era. A study showed that soaking of mites in the dsRNA solution and mites feeding on dsRNA-coated leaves were the two most efficient methods, and this delivery method may act as a mimic for sprayable dsRNA in future field applications [23].

dsRNA proved to be both (1) deliverable to *V. destructor* by adding it to the artificial diet of adult bees [24,25] or through engineering bees’ symbiontic bacteria [26] and (2) effective in controlling mites by targeting several genes. Promising results were observed in the simultaneous silencing of 14 vital genes, including those involved in the cytoskeleton, energy metabolism, and RNA polymerase, significantly reducing mite populations [24,26]. Additionally, salivary genes, which are likely critical for mite feeding, were identified as vital; their silencing corresponded with reduced mite survival [27,28]. More recently, targeting the *calmodulin* gene to disrupt *Varroa* reproduction led to a significant decrease in mite offspring, though it did not affect mite survival [25]. Despite these advances, the impact of RNAi on many biological functions of *V. destructor* remains unexplored, limiting the potential for indirect strategies such as those that interfere with the mite’s immune response.

Arthropods have a more elementary cell-mediated defense system than vertebrates because of the lack of an antigen–antibody complex and memory cells [29]. However, arthropods have evolved effective innate immunity that acts through both cellular and humoral responses directed against invaders [30]. Activation of these responses occurs through the recognition of pathogen-associated molecular patterns (PAMPs) by receptors, known as pattern recognition receptors (PRRs), located on hemocytes (immune cells) and epithelial cells from barrier sites throughout the body [31,32]. Cellular immune responses include phagocytosis, nodulation, encapsulation, and melanization events mediated by hemocytes [33,34,35]. Humoral responses orchestrated by signaling pathways such as IMD, Toll, JAK/STAT, and JNK lead to the synthesis of various defense enzymes, complement-like proteins, and antimicrobial peptides (AMPs) in response to infection [36,37,38]. Most of the knowledge about the immunity of the *Acarina* group (mites and ticks) comes from research on the immune responses of ticks, which are structured differently from those of insects and crustaceans. [39]. In ticks and other chelicerates, core intracellular components of the “canonical” IMD pathway, such as the NF-κB transcription factor Relish, are present. However, genome and functional studies indicate that ticks lack key upstream elements, including transmembrane PGRPs and the adapter molecules IMD and FADD [30,40,41,42,43].

In this study, we investigated the immune gene repertoire of *V. destructor* by querying its NCBI nr protein database. To place these genes in an evolutionary context, we conducted additional analyses, including comparisons with model species of ticks (*Ixodes scapularis*) and mites (*Galendromus occidentalis* and *Tetranychus urticae*).

## 2. Materials and Methods

We used a broadly used approach to identify immunity-related genes by inferring protein homology based on sequence similarity through BLASTp searches [44,45]. Immune gene candidates from *D. melanogaster*, reported in Table 1, Table 2 and Table 3, were used to query the NCBI nr protein database by limiting the target organism to *V. destructor*, *Ixodes scapularis*, *Galendromus occidentalis*, and *Tetranychus urticae*. Only hits with an E-Value less than 1 × 10^−5^ were considered positive.

To validate the identified candidate immune genes, we used their protein sequences to query a de novo assembled transcriptome of *V. destructor.* The transcriptome was assembled using Illumina reads obtained from a previous study (SRA BioProject ID: PRJNA1097857), in which we analyzed the salivary gland transcriptome of *V. destructor* by comparing it with the transcriptome of the rest of the adult female body [28]. First, reads of all the samples, obtained from more than 140 mites, were cleaned using Trimmomatic [46], merged, and de novo assembled with Trinity [47]. Then, the candidate immune genes were used as the query in tBLASTn searches against the assembled transcriptome. We considered positivity at the transcription level only if the query sequenced had more than 80% identity with the best hit in the transcriptome.

To reconstruct the immune gene phylogeny, putative homologous sequences in other insect species (*Apis mellifera*, *Tribolium castaneum*, and *Bombyx mori*) were identified by sequence similarity searches through BLASTp, using *D. melanogaster* proteins (Table 1, Table 2 and Table 3) as the query versus the non-redundant NCBI database (nr NCBI db). One best hit per query was selected and all the protein sequences were aligned using Muscle 3.8 [48], with default settings.

Alignments were automatically trimmed using Gblocks version 0.91b [49] to avoid comparisons of non-conserved regions present only in a subset of the taxa. The best-fit model of amino acid substitution and phylogenetic reconstruction was generated using RAxML 8.2.12 [50]. The maximum likelihood tree was run for 1000 bootstrap replicates and the tree figure was plotted using FigTree v1.4.3. Protein sequences were analyzed with ScanProsite (https://prosite.expasy.org/scanprosite/, accessed on 5 November 2024) and InterProScan [51], in order to identify active sites, conserved patterns, and domains [52]. For PGRPs, we scanned putative homologs for transmembrane helices using DeepTMHMM, which predicts alpha and beta transmembrane proteins using deep neural networks [53].

## 3. Results and Discussion

### 3.1. Overview of the Immune Gene Survey

We selected a subset of genes involved in the three phases of the arthropod immune response: recognition, signaling, and response. By using protein sequences from *D. melanogaster*, known to be involved in immunity, as queries in BLASTp searches, we successfully identified putative homologs of *Drosophila* canonical immune genes in *V. destructor*. Specifically, we identified proteins related to recognition, signaling, and response to pathogenic microorganisms (Figure 1).

### 3.2. Recognition Genes

Our analysis pointed out the occurrence of proteins encoded by the *V. destructor* genome with significant matches with *Drosophila* genes involved in recognition (Table 1). Among these, only one candidate recognition gene was not transcribed in adult female mites (Appendix A). As occurs in ticks and predatory/phytophagous mites (Appendix A), *V. destructor* lacks transmembrane peptidoglycan recognition proteins (PGRPs), Gram-negative binding proteins, and some members of the c-type lectin family. 

**Table 1 insects-16-00356-t001:** Immune-related proteins of *Varroa destructor* involved in recognition. Orthologs not found in *V. destructor* are in italics.

Gene Name	Role	*D. melanogaster*	*V. destructor* ^1^	E-Value	Identity	Coverage	Transcription
*PGRP-LC, peptidoglycan* *recognition protein*	bacterial recognition	AAF50302.3	XP_022660134.1	2 × 10^−30^	36.71%	31%	YES
*PGRP-LE, peptidoglycan* *recognition protein*	activation of PPO cascade and autophagy	NP_573078.1	XP_022660134.1	5 × 10^−25^	31.01%	45%	YES
*PGRP-SA, peptidoglycan* *recognition protein*	bacterial recognition	AAF48056.1	XP_022660135.1	9 × 10^−34^	32.93%	82%	YES
*PGRP-SD, peptidoglycan* *recognition protein*	bacterial recognition	CAD89193.1	XP_022660134.1	5 × 10^−27^	33.54%	86%	YES
*PGRP-LB, peptidoglycan* *recognition protein*	bacterial recognition	NP_650079.1	XP_022660134.1	2 × 10^−31^	35.09%	79%	YES
*PGRP-SC1a, peptidoglycan* *recognition protein*	bacterial recognition	CAD89161.1	XP_022660134.1	3 × 10^−32^	30.62%	86%	YES
*PGRP-SC2, peptidoglycan* *recognition protein*	bacterial recognition	CAD89187.1	XP_022660134.1	3 × 10^−35^	31.71%	89%	YES
*PGRP-SB1, peptidoglycan* *recognition protein*	PGN degradation and antibacterial activity	CAD89136.1	XP_022660135.1	6 × 10^−35^	35.22%	83%	YES
*PGRP-LF, peptidoglycan* *recognition protein*	blocking of IMD pathway	NP_648299.3	XP_022660134.1	3 × 10^−38^	37.35%	79%	YES
*PGRP-LA, peptidoglycan* *recognition protein*	*activation of IMD* *pathway*	*AAF50304.2*	*Not found*	*Not found*	*Not found*	*Not found*	*-*
*GNBP1, Gram-negative* *binding protein 1*	*bacterial and* *fungal pattern* *recognition*	*Q9NHB0.2*	*Not found*	*Not found*	*Not found*	*Not found*	*-*
*GNBP2, Gram-negative* *binding protein 2*	*bacterial and* *fungal pattern* *recognition*	*ACU30172.1*	*Not found*	*Not found*	*Not found*	*Not found*	*-*
*GNBP3, Gram-negative* *binding protein 3*	*bacterial and* *fungal pattern* *recognition*	*CAJ18910.1*	*Not found*	*Not found*	*Not found*	*Not found*	*-*
*DL1, c-type lectin 1*	*bacterial* *recognition*, *induction of PPO* *cascade*	*AAF53793.1*	*Not found*	*Not found*	*Not found*	*Not found*	*-*
*DL2, c-type lectin 2*	*bacterial* *recognition*, *induction of PPO* *cascade*	*NP_001014489.1*	*Not found*	*Not found*	*Not found*	*Not found*	*-*
*DL3, c-type lectin 3 or solute carrier*	bacterial recognition, induction of PPO cascade	NP_001014490.1	XP_022646715.1	4 × 10^−7^	23.58%	79%	YES
*galectin 4*	several roles have been hypothesized	ADZ99399.1	XP_022654763.1	1 × 10^−9^	34.75%	29%	YES
*TEP1, CD109 antigen-like*	mark pathogens for phagocytosis	CAB87807.1	XP_022645189.1	0.0	31.32%	98%	YES
*TEP2, CD109 antigen-like*	mark pathogens	CAB87808.1	XP_022645189.1	0.0	33.79%	98%	YES
*TEP3, CD109 antigen-like*	mark pathogens	AAL39195.1	XP_022645188.1	0.0	32.35%	97%	YES
*TEP4, CD109 antigen-like*	mark pathogens	NP_523603.2	XP_022645188.1	0.0	30.90%	98%	YES
*Pes, scavenger receptor class B member 1-like*	bacterial and fungal recognition	AHN54246.1	XP_022673026.1	4 × 10^−61^	29.15%	78%	YES
*Crq, croquemort, lysosome membrane protein 2-like*	bacterial and fungal recognition	AAF51494.1	XP_022672747.1	5 × 10^−70^	28.81%	96%	YES
*Drpr, protein draper-like*	bacterial and fungal recognition	NP_477450.1	XP_022656525.1	5 × 10^−69^	39.01%	72%	YES
*sr-CI, scavenger receptor* *class C, type i*	bind to lipoproteins and bacteria	AAW79470.1	XP_022658153.1	3 × 10^−24^	28.62%	45%	YES
*sr-CII, scavenger receptor* *class C, type ii*	bind to lipoproteins and bacteria	AAF58551.1	XP_022658154.1	4 × 10^−23^	27.16%	49%	YES
*sr-CIII, scavenger receptor* *class C, type iii*	bind to lipoproteins and bacteria	AAF37564.1	XP_022658154.1	2 × 10^−9^	21.00%	90%	YES
*sr-CIV, scavenger receptor* *class C, type iv*	bind to lipoproteins and bacteria	AAF51092.1	XP_022658154.1	5 × 10^−19^	26.59%	73%	YES
*eater*	receptor in phagocytosis and microbial binding	AAF56664.5	XP_022668925.1	2 × 10^−17^	33.72%	77%	NO
*Drp, protein draper-like*	receptor in phagocytosis and microbial binding	AAF53364.2	XP_022656529.1	6 × 10^−12^	27.66%	62%	YES

^1^ Best hit on the E-Value basis obtained by BLASTp searches against the NCBI nr database.

#### 3.2.1. Peptidoglycan Receptor Proteins

Nearly all bacteria have peptidoglycans as crucial components of the cell wall, which are detected by the immune system thanks to pathogen recognition receptors (PRRs). Various families of pattern recognition molecules that identify peptidoglycans have been discovered in insects, and the function of peptidoglycan recognition proteins (PGRPs) in immune defense is relatively well understood in *Drosophila* [54]. Recognition through PGRPs triggers both the Toll and IMD/JNK signaling pathways, resulting in the activation of prophenoloxidase (proPO) or the production of antimicrobial peptides [55]. Most insect species have multiple PGRP genes that vary in both structure and function. For instance, *Drosophila* contains 13 PGRP genes that produce 19 different proteins, whereas *Anopheles gambiae* has 7 PGRP genes that encode 9 distinct proteins [54]. Conversely, our analysis revealed a limited set of PGRPs, identifying only a single putative homolog gene in the species we studied (Figure 2). This gene encodes two isoforms in *V. destructor* and *I. scapularis*, and a single protein in *G. occidentalis* and *T. urticae*. Furthermore, TMHMM analysis indicated that the PGRPs in *V. destructor* lack transmembrane domains, consistent with observations in other mites and ticks [30].

This lack of transmembrane PGPRs in *Acarina* seems to be associated with the evolution of alternative protein sensors. Indeed, previous works demonstrated that the IMD and JNK pathways of the tick *I. scapularis* are activated by lipid molecular patterns [43], such as 1-palmitoyl-2-oleoyl-sn-glycero-phosphoglycerol (POPG), sensed by the homolog of croquemort (Crq) [39], a CD36-like protein originally identified in *Drosophila* [56]. Here, we report the presence of the Crq homolog in *V. destructor* (Table 1), which suggests a similar mechanism of recognition in mites.

#### 3.2.2. Gram-Negative Binding Proteins

The structure of Gram-negative binding proteins (GNBPs) is characterized by a carbohydrate-binding module (CBM) at the N-terminus and a glucanase-like domain (Glu) at the C-terminus [57]. The CBM engages with microbial polysaccharides, whereas the Glu domain interacts with subsequent proteases, thus triggering immune signaling pathways [58]. GNBPs detect both bacterial and fungal pathogens, thus activating immune signaling cascades in insects [59]. In *Drosophila*, GNBP1 and peptidoglycan-recognition protein SA (PGRP-SA) work together to trigger the Toll pathway in response to Gram-positive bacterial infections [60], while GNBP3 plays a crucial role in activating the Toll pathway during fungal infections [61].

The comparative study of Palmer and Jiggins (2015) revealed that GNBPs, also known as βGRPs, have been lost from *chelicerates*, likely through a single-loss event [30]. *V. destructor* is no exception and, like other *Acarina* species (Appendix A), shows no similarities to known GNBPs of *D. melanogaster* (Table 1).

#### 3.2.3. Lectins

Lectins, a diverse class of carbohydrate-binding proteins, play a crucial role in the immune defense of numerous insect species. These proteins are recognized for their wide-ranging ability to bind pathogens and their participation in various immune functions, including opsonization, melanization, synthesis of antibacterial peptides, encapsulation, and the direct elimination of bacteria [62,63]. The *Drosophila* c-type lectins (CTLs) DL2 and DL3 agglutinate Gram-negative *Escherichia coli* in a calcium-dependent manner [64]. Our analysis pointed out a reduced CTL repertoire in *Acarina*. *V. destructor* lacks homologs of DL1 and DL2 but possesses a putative homolog of DL3 (Table 1). Notably, no CTLs were detected in the genomes of *I. scapularis*, *G. occidentalis*, and *T. urticae* when using *Drosophila* sequences as the query (Appendix A). However, by using the DL3 putative homolog identified in *V. destructor* as the query, significant hits were obtained for both *I. scapularis* and *G. occidentalis*, though not for *T. urticae* (Appendix A), where the best match was with a sushi, von Willebrand factor type A, EGF, and pentraxin domain-containing protein 1. This intriguing reduction in CTLs in *T. urticae* is in line with what was observed after immune challenge in this phytophagous mite by bacteria injection. Indeed, *T. urticae* has lost the capacity to mount an induced immune response against bacteria, in contrast to other mites and chelicerates but similarly to the phloem-feeding aphid *Acyrthosiphon pisum* [65]. The suggested evolutionary link between ecological conditions regarding exposure to bacteria and the architecture of the immune response deserves further studies. Although DL3 is predicted to be a secreted protein, it is detected on the surface of *Drosophila* hemocytes and enhances encapsulation [64]. Similarly, our in silico analysis suggests that the DL3 homolog identified in *V. destructor* is also predicted to be secreted, indicating a potential analogous role in mite immunity.

Galectins, a widely prevalent family of lectins [66], exhibiting thiol group-dependent activity and present in both intracellular and extracellular spaces [67], are upregulated in mosquitoes upon exposure to bacterial infections as well as malaria parasite invasion [68]. Galectins present in the gut of the hematophagous sand fly *Phlebotomus papatasi* facilitate a specific interaction between the insect’s midgut and *Leishmania major*, a critical event for parasite survival, and play a role in determining species-specific vector competence [69]. More broadly, insect galectins are believed to participate in pathogen recognition, agglutination, and phagocytosis [66,70]. Variability in galectin transcripts across different insect species has been highlighted by genome-wide studies: 5 transcripts were identified in *Drosophila melanogaster*, 8 in *Anopheles gambiae*, and 12 in *Aedes aegypti* [71], with 4 identified in several *Lepidoptera* species [72]. *Acarina* species (Appendix A), including *V. destructor*, possess putative homologs of galectins (Table 1). Notably, *I. scapularis* shows a significant expansion of galectins, compared to the other target species, which have diversified into three distinct clusters (Figure 3).

#### 3.2.4. Thioester-Containing Proteins

Thioester-containing proteins (TEPs) are a family of proteins that show similarities to vertebrate alpha-2 macroglobulins (A2M) and complement factors [73,74]. Seven members of this family are encoded in the human genome: C3, C4, C5, A2M, pregnancy zone protein (PZP), CD109, and the complement 3 and PZP-like A2M domain-containing 8 (CPAMD8) [75,76,77]. As seen in complement proteins, certain insect TEPs play a role in the opsonization of microbes and pathogens, effectively “labeling” them for phagocytosis, melanization, and the formation of lytic complexes [78,79]. Given their participation in microbe recognition, TEPs are categorized as pattern recognition receptors (PRRs) [80].

Our phylogenetic analysis revealed a distinct clustering of A2M, CD109, and C3 homologs (Figure 4). All *Acarina* species investigated had at least one representative in each cluster, except for *T. urticae*, which lacks putative homologs of A2M. Notably, the best BLASTp hits of *Drosophila* TEP1, TEP2, TEP3, and TEP4 in the *V. destructor* genome are two isoforms of the same gene (Table 1).

### 3.3. Signaling Pathways

Our analysis revealed the occurrence of protein sequences encoded by the *V. destructor* genome with significant matches with genes of *Drosophila* involved in signaling. Among these, four candidate signaling genes were not transcribed in adult female mites (Appendix A). As occurs in other mites and ticks (Appendix A), the *V. destructor* genome includes a complete Toll pathway, while it lacks some members of IMD (IMD, dFADD, TAB2), JAK/STAT (unpaired), and JNK (Eiger) signaling pathways, which are present in *Drosophila* genomes (Table 2).

**Table 2 insects-16-00356-t002:** Immune genes of *Varroa destructor* involved in signaling. Orthologs not found in *V. destructor* are in italics.

Gene Name	Role	*D. melanogaster*	*V. destructor* ^1^	E-Value	Identity	Coverage	Transcript
*Spz1-1, spätzle 1B*	Toll pathway	NP_733188.1	XP_022644257.1	6 × 10^−7^	25.32%	62%	NO
*Spz1-2, spätzle 1Bii*	Toll pathway	NP_001138116.1	XP_022644257.1	2 × 10^−7^	25.32%	52%	NO
*Spz2, spätzle 2*, *neurotrophin 1*	Toll pathway	NP_001261417.1	XP_022654810.1	3 × 10^−15^	27.85%	14%	YES
*Spz3, spätzle 3*	Toll pathway	NP_609160.2	XP_022643975.1	3 × 10^−61^	40.00%	43%	NO
*Spz4, spätzle 4*	Toll pathway	NP_609504.2	XP_022644415.1	5 × 10^−51^	63.55%	17%	YES
*Spz5, spätzle 5*	Toll pathway	NP_647753.1	XP_022663991.1	1 × 10^−14^	38.98%	29%	YES
*Spz6, spätzle 6*	Toll pathway	NP_611961.1	XP_022645393.1	2 × 10^−30^	63.75%	18%	YES
*Toll-1, protein Toll*	Toll pathway	NP_524518.1	XP_022664896.1	3 × 10^−93^	29.56%	74%	YES
*18 wheeler, Toll-2*	Toll pathway	NP_476814.1	XP_022651559.1	0.0	38.74%	83%	YES
*Toll-6*	Toll pathway	NP_001246766.1	XP_022653722.1	0.0	40.50%	77%	NO
*Toll-7*	Toll pathway	NP_523797.1	XP_022656113.1	0.0	41.21%	82%	YES
*Tollo, Toll-8*	Toll pathway	NP_524757.1	XP_022651559.1	0.0	40.47%	89%	YES
*Tube ^2^, interleukin-1* *receptor-associated* *kinase 4*	Toll pathway	NP_001189164.1	Not found. A putative homolog was found by using the query from the *T. urticae* genome ^2^.	Not found	Not found	Not found	-
*Myd88, myeloid* *differentiation primary* *response gene ^2^*	Toll pathway	AAF58953.1	Not found. A putative homolog was found by using the query from the *T. urticae* genome ^2^.	Not found	Not found	Not found	-
*Pll, pelle*	Toll pathway	AAF56686.1	XP_022652064.1	1 × 10^−64^	42.48%	60%	YES
*Cact, cactus*	Toll pathway	AAN10936.1	XP_022667531.1	8 × 10^−17^	30.21%	48%	YES
*Cactin*	Toll pathway	NP_523422.4	XP_022668977.1	0.0	52.09%	82%	YES
*Pli, pellino*	Toll pathway	NP_524466.1	XP_022668132.1	8 × 10^−68^	42.61%	59%	YES
*Traf1, TNF-receptor-associated factor 1*	Toll pathway	AAD34346.1	XP_022668810.1	8 × 10^−101^	46.20%	66%	YES
*Traf2,TNF-receptor-associated factor 2*	Toll pathway	AAF46338.1	XP_022668812.1	7 × 10^−9^	24.84%	30%	YES
*Traf3, TNF-receptor-associated factor 3*	Toll pathway	NP_727976.1	XP_022687269.1	2 × 10^−15^	25.00%	38%	YES
*Dl, dorsal*	Toll pathway	AAF53611.1	XP_022665229.1	8 × 10^−87^	50.37%	39%	YES
*Dome, domeless 1*, *interleukine* *JAK/STAT receptor*	JAK/STAT pathway	CAD12503.1	XP_022659768.1	2 × 10^−21^	21.79%	39%	YES
*Hops, hopscotch, Janus* *kinase*	JAK/STAT pathway	NP_511119.2	XP_022650171.1	2 × 10^−52^	42.59%	61%	YES
*STAT92E, signal transducer and* *activator of* *transcription, marelle*	JAK/STAT pathway	AAX33462.1	XP_022661521.1	1 × 10^−129^	36.85%	94%	YES
*unpaired 1*	*JAK/STAT pathway*	*NP_525095.2*	*Not found*	*Not found*	*Not found*	*Not found*	*-*
*unpaired 2*	*JAK/STAT pathway*	*NP_001356882.1*	*Not found*	*Not found*	*Not found*	*Not found*	*-*
*unpaired 3*	*JAK/STAT pathway*	*NP_001097014.1*	*Not found*	*Not found*	*Not found*	*Not found*	*-*
*IMD, immune deficiency*	*IMD pathway*	*NP_573394.1*	*Not found*	*Not found*	*Not found*	*Not found*	*-*
*dFADD*	*IMD pathway*	*NP_651006.1*	*Not found*	*Not found*	*Not found*	*Not found*	*-*
*Dredd, death-related ced-3*, *caspase-1*	IMD pathway	NP_477249.3	XP_022672682.1	2 × 10^−17^	28.98%	46%	YES
*Rel, Relish ^3^*	IMD pathway	NP_477094.1	XP_022656314.1	3 × 10^−11^	33.81%	14%	YES
*TAB2, TAK1-associated* *binding protein 2*	*IMD pathway*	*NP_611408.2*	*Not found*	*Not found*	*Not found*	*Not found*	*-*
*TAK1, TGF-β-activated* *kinase 1*	IMD pathway	AAF50895.1	XP_022652311.1	8 × 10^−81^	41.39%	64%	YES
*Key, Kenny*	*IMD pathway*	*NP_523856.2*	*Not found*	*Not found*	*Not found*	*Not found*	*-*
*DIAP2/XIAP, death-associated* *inhibitor of apoptosis 2*	IMD pathway	NP_477127.1	XP_022671835.1	2 × 10^−59^	31.12%	86%	YES
*IRD5, immune response* *deficiency 5, IK-β*, *IKKB, I-kappa-B kinase* *beta*	IMD pathway	NP_524751.3	XP_022693386.1	9 × 10^−37^	38.05%	31%	YES
*Hep, hemipterous*	JNK pathway	NP_727661.1	XP_022657489.1	4 × 10^−110^	56.25%	25%	YES
*Bsk, basket*	JNK pathway	P92208.1	XP_022646497.1	0.0	84.72%	96%	YES
*Jra, Jun-related antigen*	JNK pathway	AAF58845.1	XP_022649508.1	2 × 10^−32^	34.36%	78%	YES
*Kay, kayak*	JNK pathway	NP_001027579.1	XP_022661416.1	2 × 10^−10^	41.44%	23%	YES
*Egr, Eiger*	*JNK pathway*	*AAF58848.2*	*Not found*	*Not found*	*Not found*	*Not found*	*-*

^1^ Best hit on the E-Value basis obtained by BLASTp searches against the NCBI nr database. ^2^ See Appendix A. ^3^ This sequence does not contain Relish domains and it cannot be considered a Relish homolog (see Section 3.3.3).

#### 3.3.1. The Toll Signaling Pathway

The Toll pathway in *Drosophila* plays a pivotal role in both development and innate immunity. The deletion of its constituent genes enhances susceptibility to a range of pathogens, including Gram-positive bacteria, fungal infections, certain Gram-negative bacteria, and viruses [81]. The Toll pathway appears to be intact in *V. destructor* (Table 2) and other *Acarina* species (Appendix A), according to other studies on *chelicerates* immunity-related genes [30,41]. We found convincing matches for genes encoding the extracellular cytokine spätzle, the transmembrane receptor Toll (Figure 5), the kinase pelle, cactin, pellino, Traf, and the transactivator dorsal (Table 2).

Moreover, our findings include a putative homolog of MyD88 and the *tube* adaptor in the *Acarina* species studied (Appendix A). In invertebrates, MyD88 together with the adaptor tube form a complex with the kinase pelle, which in turn leads to the activation of cactus, and ultimately to the activation of the NF-κB ortholog dorsal [82]. Notably, Palmer and Jiggins, in their comparative genomics study, failed to find a tube homolog in any of the species studied, suggesting that the cause might be a lack of the power needed to detect the gene [30]. Indeed, in another comparative study the identification of the tube orthologs in arachnids was possible only by using the tube sequence from the mosquito *A. aegypti*, while no matches were obtained using the *Drosophila* ortholog as the query [41]. Similarly, we found that BLASTp searches using the *Drosophila* tube sequence as the query revealed no hits in *V. destructor* (Table 1), *I. scapularis* (Appendix A), and *G. occidentalis* (Appendix A) genomes, while a significant match was found in *T. urticae* (Appendix A). Using this sequence as an alternative query, we identified significant matches in our target species, suggesting the presence of a tube homolog in the genomes of *Acarina*, including *V. destructor* (Appendix A). However, tube and pelle are members of the same gene family, both having a death domain and a kinase domain, and orthology is difficult to infer.

*Acarina* also possess multiple genes encoding Toll receptors (Figure 5), which function as transmembrane receptors in both vertebrates and invertebrates [83,84]. While nine single-copy Toll genes have been identified in *D. melanogaster* (Toll-1 to Toll-9), it appears that Acarina, like other arthropods, lack some of these genes but have multiple isoforms of others. Indeed, *V. destructor* exhibited a highly diversified repertoire of Toll putative homologs (Figure 5). Toll-like sequences in *Acarina* form monophyletic groups (Figure 5), suggesting that diversification of the Toll family occurred after the evolution of the separate lineages of *Acarina* and insects. This is in line with other phylogenetic analyses, which indicated that Toll receptor genes from different phyla fall into separate clusters, showing that they share a common ancestor but evolved independently by gene duplication [85,86]. Our results suggest a functional divergence between *Acarina* and insect Toll receptors, which may be disclosed only by functional data.

#### 3.3.2. The JAK/STAT Signaling Pathway

In *Drosophila*, the JAK/STAT pathway, much like the Toll pathway, plays pivotal roles in both development and immune responses. Although it remains the least detailed among the core insect immune pathways, it appears to be involved in hemocyte overproliferation and antiviral defense [87]. Similar to other *Acarina* species (Appendix A), *V. destructor* possesses homologs of all core JAK/STAT genes, including those encoding the cytokine receptor domeless, JAK tyrosine kinase (also referred to as hopscotch), and the STAT92E transcription factor (Table 2). However, homologs for upd (unpaired), a crucial ligand for JAK/STAT activation in *Drosophila*, were not identified. This ligand is also absent in other insects, such as *A. mellifera* [88] and *Coccidae* [45]. The presence of the core JAK/STAT components (Table 2) suggests that the JAK/STAT pathway remains functional in *Acarina*, activated by an as-yet-unidentified ligand.

#### 3.3.3. IMD and JNK Signaling Pathways

The immune deficiency (IMD) pathway is critical for fighting Gram-negative bacteria in *Drosophila* [81], and IMD pathway member knockouts influence the susceptibility to some Gram-positive bacteria and fungi as well [89]. In *Drosophila*, the pathway-initiating receptors, PGRPs, engage the adapter proteins IMD and FADD (Fas-associated protein with death domain) [90,91], with FADD subsequently interacting with Dredd (caspase-8) [92], which in turn cleaves IMD. The K63 polyubiquitylation of IMD is facilitated by the E3 ubiquitin ligase IAP2 (inhibitor of apoptosis 2) and the E2-conjugating enzymes Bendless, Uev1a, and Effete [93,94,95]. This facilitates the binding of Kenny to the inhibitor of NF-κB kinase (IKK)β [92,96]. Additionally, Dredd mediates the cleavage of Relish, leading to the nuclear translocation of its N-terminal fragment, which in turn triggers the expression of antimicrobial peptides (AMPs) [95,97,98,99]. The resulting signaling scaffold leads to cleavage of the NF-κB signaling molecule Relish, which translocates to the nucleus and promotes antimicrobial peptide (AMP) expression [93,95].

As in other *Acarina* species [100], *V. destructor* appears to be missing many crucial components of the IMD signaling pathway, such as IMD, dFADD, Kenny, and TAB2 (Table 2). However, the Relish search resulted in a positive hit characterized by only 14% coverage (Table 2), which was considered low. Indeed, InterProScan analysis revealed that similarities between the Relish gene in *D. melanogaster* and its putative homolog in *V. destructor* (XP_022656314.1) are only due to the Ankyrin domain’s presence in both sequences, while the *V. destructor* sequence lacks canonical Relish domains: the NF-kappa-B/Rel/dorsal domain (IPR011539) and Rel homology dimerization domain (IPR032397) [101,102]. However, the putative Relish homolog identified in I. scapularis includes Relish domains and was used as an alternative query to scan the *V. destructor* genome. This led to the identification of a significantly similar sequence described as proto-oncogene c-Rel-like (XP_022665225.1), showing Relish domains and lacking the Ankyrin domain, that is believed to inhibit its own nuclear translocation [97,102]. However, this loss of the Ankyrin domain, which occurs also in *I. scapularis*, *G. occidentalis*, and *T. urticae*, seems to not alter the functionality of Relish in ticks [103,104,105].

Using the Dredd protein sequence to search the *V. destructor* genome via BLASTp resulted in a best hit with 46% coverage (Table 2). However, this sequence cannot be considered a true homolog of Dredd (caspase-8) because it lacks the Dredd_N (IPR056259) and Dredd_2nd (IPR056260) domains (Figure 6). These findings align with previous studies reporting the absence of Dredd homologs in ticks and mites [41]. Dredd normally cleaves the loop sequence between the REL and Ankyrin domains of phosphorylated Relish [98,106]. Our results suggest that, in *V. destructor* and *Acarina* in general, the absence of Dredd homologs is likely due to reduced selective pressure on this caspase, stemming from the loss of its substrate: the Ankyrin domain in Relish homologs.

Despite the absence of upstream regulators, core IMD signaling molecules are active against infection, as reported by several studies on tick IMD pathways [107], which rely on Bendless, Effete, Uev1a, and XIAP (X-linked inhibitor of apoptosis), the homolog of IAP2 [43,108]. Our phylogenetic analysis indicated that all model *Acarina* species studied here, including *V. destructor*, possess homolog genes of the three ubiquitination-conjugating enzymes (UCEs): Bendless, Effete, and Uev1a (Figure 7).

In ticks, once activated, XIAP interacts with the heterodimers Bendless and Uev1a, leading to the ubiquitination of p47 in a K63-dependent manner [43,109]. Ubiquitylated p47 connects to Kenny and induces the phosphorylation of IRD5 and Relish [100]. This mechanism seems well conserved in Acarina. Indeed, we identified putative homologs of XIAP (Table 2) and p47 (Appendix A) in our analysis.

Following the initiation of IMD signaling in *Drosophila*, various proteins are recruited to the molecular scaffold within the activated cell, including transforming growth factor-β-activated kinase 1 (TAK1). TAK1, which we identified also in *V. destructor* (Table 2), promotes activation of JNK signaling in parallel to NF-κB signaling [95,110].

In *Drosophila*, the JNK pathway is involved in various developmental processes, along with wound healing, and has been suggested to regulate antimicrobial peptide gene expression and cellular immune responses [81]. The putative homologs identified in *V. destructor* include kay (kayak), hep (hemipterous), bsk (basket), and jra (Jun-related antigen) (Table 2). Searches for homologs to the *Drosophila* Eiger gene found no hits in *Acarina* (Appendix A), including *V. destructor* (Table 2), as observed also for other phyla such as *Coccidae* (*Insecta, Hemiptera*) [45].

### 3.4. Response Genes

Our analysis revealed the occurrence of several proteins encoded by the *V. destructor* genome with significant matches with genes of *Drosophila* involved in the immune response. Among these, only two candidate response genes were not transcribed in adult female mites (Appendix A). As occurs in ticks and mites (Appendix A), *V. destructor* lacks canonical antimicrobial peptides and prophenoloxidase orthologs, which are present in the *D. melanogaster* genome (Table 3).

**Table 3 insects-16-00356-t003:** Immune genes of *Varroa destructor* involved in the immune response. Orthologs not found in *V. destructor* are in italics.

Gene Name	Role	*D. melanogaster*	*V. destructor* ^1^	E-Value	Identity	Coverage	Transcript
*Att, Attacin*	*antimicrobial* *peptide*	*NP_523745.1*	*Not found*	*Not found*	*Not found*	*Not found*	*-*
*Cec, Cecropin*	*antimicrobial* *peptide*	*C0HKQ7.1*	*Not found*	*Not found*	*Not found*	*Not found*	*-*
*Def, Defensin*	*antimicrobial* *peptide*	*ANY27112.1*	*Not found*	*Not found*	*Not found*	*Not found*	*-*
*Dro, Dosocin*	*antimicrobial* *peptide*	*XP_016946682.1*	*Not found*	*Not found*	*Not found*	*Not found*	*-*
*Mtk, Metchnikowin*	*antimicrobial* *peptide*	*AAO72489.1*	*Not found*	*Not found*	*Not found*	*Not found*	*-*
*Andropin*	*antimicrobial* *peptide*	*P21663.1*	*Not found*	*Not found*	*Not found*	*Not found*	*-*
*Diptericin*	*antimicrobial* *peptide*	*QER92349.1*	*Not found*	*Not found*	*Not found*	*Not found*	*-*
*Drs, Drosomycin*	*antimicrobial* *peptide*	*ANY27466.1*	*Not found*	*Not found*	*Not found*	*Not found*	*-*
*LysX, Lysozyme X, i-type*	microbial degradation	CAL85493.1	XP_022671742.1	2 × 10^−30^	38.30%	96%	YES
*LysB, Lysozyme B, i-type*	microbial degradation	NP_001261245.1	XP_022671739.1	2 × 10^−33^	41.01%	97%	YES
*LysP, Lysozyme, i-type*	microbial degradation	NP_476828.1	XP_022643422.1	4 × 10^−37^	43.28%	95%	NO
*LysE, Lysozyme E*	microbial degradation	CAA80228	XP_022669180.1	3 × 10^−32^	43.09%	87%	YES
*LysD, Lysozyme D*	microbial degradation	NP_476823.1	XP_022671742.1	7 × 10^−33^	43.90%	87%	YES
*LysE, Lysozyme E*	microbial degradation	NP_476827.2	XP_022671742.1	6 × 10^−32^	39.42%	97%	YES
*LysS, Lysozyme S*	microbial degradation	NP_476829.1	XP_022671742.1	1 × 10^−34^	44.36%	95%	YES
*Lysozyme E, i-type*	microbial degradation	ACD99447.1	XP_022648598.1	5 × 10^−23^	34.55%	89%	YES
*Lysozyme, i-type*	microbial degradation	NP_611164.3	XP_022655568.1	6 × 10^−19^	35.17%	88%	YES
*Lysozyme, i-type*	microbial degradation	NP_611163.2	XP_022644325.1	4 × 10^−19^	37.30%	73%	YES
*Cht2, Chitinase-like* *protein 4*, *flocculation* *protein*	fungal degradation	NP_001261282.1	XP_022662471.1	2 × 10^−85^	40.37%	73%	YES
*Cht4, Chitinase-like* *protein 2, mucin*	fungal degradation	NP_524962.2	XP_022662425.1	9 × 10^−112^	38.54%	94%	YES
*Cht5, Chitinase-like* *protein 5*, *endochitinase*	fungal degradation	NP_650314.1	XP_022664603.1	7 × 10^−165^	43.43%	95%	YES
*Cht7, Chitinase-like* *protein 7, chitinase* *10*	fungal degradation	NP_647768.3	XP_022669697.1	0.0	52.05%	99%	NO
*Cht6, Chitinase 6*	fungal degradation	NP_001245599.1	XP_022662476.1	0.0	49.73%	47%	YES
*idgf6*	fungal degradation	NP_001286499.1	XP_022662425.1	1 × 10^−45^	27.81%	98%	YES
*PPO1, Prophenoloxidase* *1*	*prophenoloxidase* *response*	*NP_476812.1*	*Not found*	*Not found*	*Not found*	*Not found*	*-*
*PPO2, Prophenoloxidase* *2*	*prophenoloxidase* *response*	*NP_610443.1*	*Not found*	*Not found*	*Not found*	*Not found*	*-*
*PAF2, Phenoloxidase-activating factor 2*	phenoloxidase activation	AAO24923.1	XP_022659018.1	2 × 10^−82^	41.99%	77%	YES
*SP, Serine* *protease-like* *precursor*	phenoloxidase activation	NP_001097766.1	XP_022662993.1	2 × 10^−43^	37.84%	56%	YES
*Hmct, Hemolectin, hemocytin*	cell aggregation	NP_001261809.1	XP_022654008.1	8 × 10^−158^	32.23%	66%	YES
*Nos, Nitric oxide* *synthase*	production of nitric oxide	NP_001027243.2	XP_022665384.1	6 × 10^−68^	29.27%	60%	YES
*Tg, Transglutaminase*	clotting	NP_609174.1	XP_022666443.1	9 × 10^−121^	33.06%	90%	YES

^1^ Best hit on the E-Value basis obtained by BLASTp searches against the NCBI nr database.

#### 3.4.1. Antimicrobial Peptides

Antimicrobial peptides (AMPs) are pivotal components of the immune defenses across a broad range of organisms, including insects. These peptides are among the most extensively researched humoral effectors and can be produced either constitutively or in response to specific signaling pathways. Typically composed of 15-to-50 amino acids, AMPs possess an amphipathic structure that enables them to disrupt microbial membrane integrity, thereby inducing alterations in the osmotic equilibrium and ultimately leading to lysis. The activation of the IMD and Toll signaling pathways in *Drosophila* triggers the transcription of distinct antimicrobial peptides [111]. At present, numerous structural families of insect-derived AMPs are identified, such as defensins, cecropins, drosocins, attacins, diptericins, metchnikowins, and melittins, some of which are exclusive to certain taxonomic groups [112]. However, none of these were identified in *V. destructor* and other *Acarina* species studied here by our BLASTp searches (Table 3). In contrast to our findings, defensins were reported from several tick species, including *I. scapularis* [113], suggesting that BLASTp is not the most appropriate method to identify these antimicrobial peptides in our target species. Indeed, we used the defensins identified in *I. scapularis* as alternative queries to scan the *V. destructor* proteome, which again resulted in no significant match. This limitation of BLASTp searches aimed at identifying AMPs was observed by other similar studies. As suggested, the small size of AMPs makes their identification by sequence similarity problematic across large phylogenetic distances [30].

Ctenidins are a family of constitutively expressed, glycine-rich AMPs identified from the spider *Cupiennius salei* that exhibit activity against Gram-negative bacteria [114]. BLASTp using ctenidin-like peptides identified in ticks as queries to scan the *V. destructor* genome led to the identification of putative homologs (Appendix A). A similar pattern is observed for *G. occidentalis*, but not in *T. urticae*, where ctenidin putative homologs were not found (Appendix A). These findings are particularly significant given the absence of widely adopted genetic readouts for immune pathways in mites [39].

#### 3.4.2. Lysozyme

Lysozymes, enzymes tasked with degrading bacterial cell walls through the cleavage of the polysaccharide component of peptidoglycan, are categorized into two distinct classes in insects: the c-type, which exhibits muramidase activity, and the i-type, which possesses both muramidase and isopeptidase activities [115,116].

In ticks, lysozymes are expressed in the gut, salivary glands, and hemolymph, and are upregulated in response to bacterial challenge, suggesting a possible role of lysozymes: giving innate immunity to ticks against microorganisms [117]. Our analysis identified seven different proteins of *V. destructor* with significant similarity to *Drosophila* lysozymes (Table 3). Among these, only three (XP_022648598, XP_022655568, and XP_022644325) contain the invertebrate (I)-type lysozyme domain profile (LYSOZYME_I). Two of these (XP_022655568 and XP_022644325) also carry the EF hand calcium-binding domain (EF_HAND_1).

By filtering BLASTp results using the same criteria, we identified three lysozyme homologs in *G. occidentalis* and two in *I. scapularis*, carrying both LYSOZYME_I and EF_HAND_1. ScanProsite analysis revealed that among the sequences identified in *T. urticae*, large proteins (>400AA) possess multiple LYSOZYME_Is, while only one sequence (XP_015789674) showed a single LYSOZYME_I and was in the length range of canonical lysozymes (176AA). All these sequences identified in *Acarina* form a monophyletic group that diverges from the lysozymes with LYSOZYME_I and EF_HAND_1 identified in insect species (Figure 8).

#### 3.4.3. Melanization and Coagulation

Melanization, a key immune response in arthropods, involves the encapsulation of pathogens with melanin and their subsequent elimination through toxic quinone precursors [55,118]. Central to this melanization process is the enzyme prophenoloxidase (proPO, PPO), which, upon activation, transforms ortho-diphenols such as dopamine into quinones, precursors to melanin that ultimately encase and neutralize the pathogen. It is believed that in the common ancestor of arthropods, the oxygen-carrying molecule hemocyanin likely evolved from a protein resembling PO [119]. While PO has been detected in insects and crustaceans, specific forms of PO have not been identified in the studied chelicerates. Notably, in the horseshoe crab *T. tridentatus*, the oxygen-transporting protein hemocyanin exhibits PO activity upon activation [120]. Similarly, a hemocyanin with PO activity has been observed in the tarantula species *Eurypelma californicum* [121].

There is no evidence of the existence of the PPO cascade in ticks based on the available genomic and transcriptomic data [100]. In line with this, no PPO activity has been reported to be present in the hemolymph of the hard ticks *Amblyomma americanum*, *Dermacentor variabilis*, and *I. scapularis* [122]. Moreover, ticks lack coagulin and hemocyanin, suggesting a unique coagulation system in *Acarina*, which is largely different from that of insects [123]. Our analysis showed that this pattern occurs also in mites, including *V. destructor*, which lack PPO but possess transglutaminase homologs (Table 3). Considering these findings, additional studies are required to characterize melanization and the linked coagulation responses in *Acarina*.

## 4. Conclusions

Our results shed light on the genetic basis of *Varroa* and *Acarina* immunity, which is important for understanding the evolution of immune response in an economically important group of arthropods. While the genomic resources obtained through different sequencing technologies may result in different genome coverage, introducing bias in comparative analysis, we observed a clear pattern of immune gene loss in *V. destructor*, and chelicerates in general, supported by previous studies on arthropod immunity. Indeed, the immune gene repertoire of *V. destructor* matches the same pattern that several studies report for ticks, including the lack of transmembrane peptidoglycan recognition proteins (PGRPs), Gram-negative binding proteins, and several lectin families involved in recognition. Regarding signaling pathways, as occurs in ticks, *Varroa* mites lack homologs of the unpaired ligand (JAK/STAT), Eiger (JNK), and many components of IMD. *Varroa* mites also lack homologs of canonical AMPs, such as defensins, while their genome encodes putative homologs of ctenidins, AMPs previously identified in spiders and ticks. The same pattern is consistent in the predatory mite *G. occidentalis*, but not conserved in the phytophagous mite *T. urticae*, for which no canonical lectins nor ctenidins were identified. This suggests a greater degree of immune gene erosion in herbivorous mites, a trend that warrants further investigation.

Although our findings stem from a conservative homology-based approach using canonical insect immune genes, they represent the first comprehensive checklist of immunity-related genes in *V. destructor*. We hope this initial overview will inspire future studies to focus on the functional characterization of these genes, an essential step toward studying immune response in mites and developing novel strategies for pest control based on immunosuppression.

## Figures and Tables

**Figure 1 insects-16-00356-f001:**
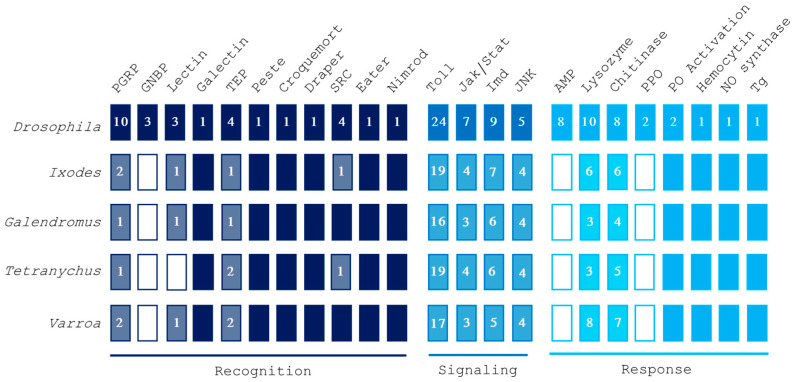
Overview of gene families involved in recognition, signaling, and response pathways in *Varroa destructor*, *Tetranychus urticae, Galendromus occidentalis*, and *Ixodes scapularis.* Putative homology was inferred by using *Drosophila melanogaster* queries. Numbers in blocks indicate the amount of different genes identified for each protein family. Filled boxes imply the presence of all the analyzed genes, while empty boxes represent a total absence of them. Abbreviations: peptidoglycan recognition protein (PGRP), Gram-negative binding protein (GNBP), thioester-containing protein (TEP), scavenger receptor class C (SRC), antimicrobial peptide (AMP), pro-phenoloxidase (PPO), phenoloxidase (PO), nitric oxide (NO), transglutaminase (Tg).

**Figure 2 insects-16-00356-f002:**
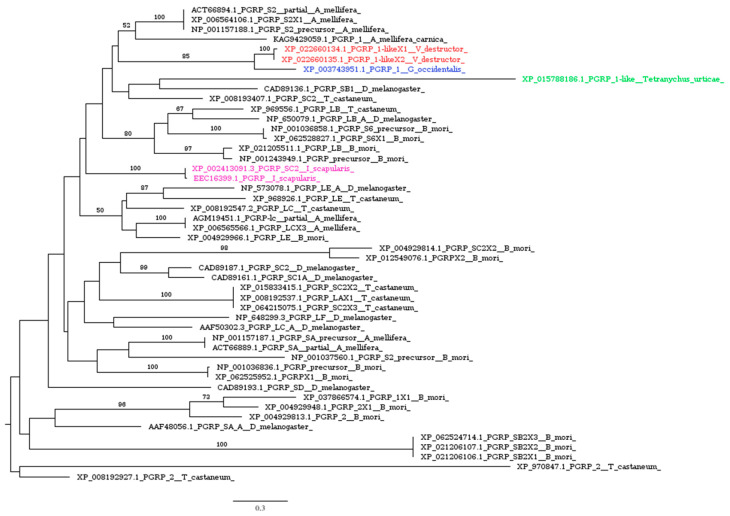
Phylogenetic tree based on maximum likelihood analysis of PGRP putative homologs identified in the genomes of *V. destructor* (in red), *G. occidentalis* (in blue), *T. urticae* (in green), and *I. scapularis* (in purple). The longest branch of the unrooted tree is used as the outgroup. Bootstrap support values > 50% are indicated at each node. The scale bar indicates the number of amino acid substitutions per site.

**Figure 3 insects-16-00356-f003:**
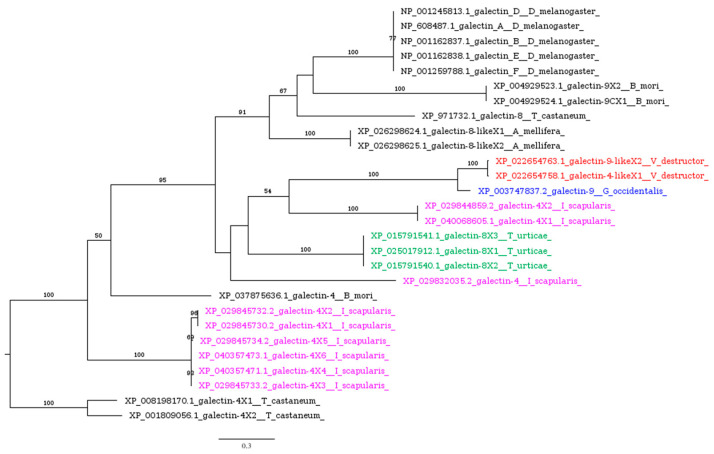
Phylogenetic tree based on maximum likelihood analysis of galectin putative homologs identified in the genomes of *V. destructor* (in red), *G. occidentalis* (in blue), *T. urticae* (in green), and *I. scapularis* (in purple). The longest branch of the unrooted tree is used as the outgroup. Bootstrap support values >50% are indicated at each node. The scale bar indicates the number of amino acid substitutions per site.

**Figure 4 insects-16-00356-f004:**
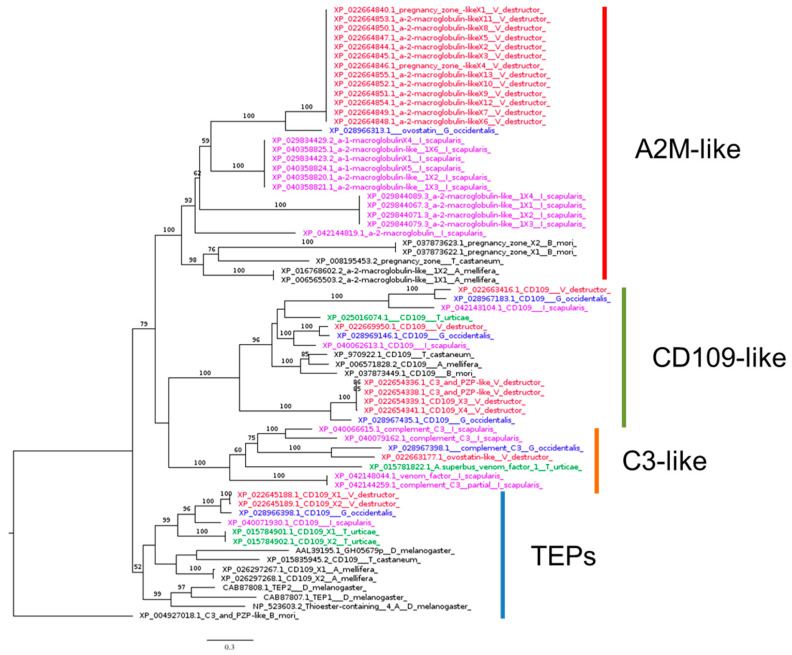
Phylogenetic tree based on maximum likelihood analysis of thioester-containing protein (TEP) identified in the genomes of *V. destructor* (in red), *G. occidentalis* (in blue), *T. urticae* (in green), and *I. scapularis* (in purple). The longest branch of the unrooted tree is used as the outgroup. Bootstrap support values > 50% are indicated at each node. The scale bar indicates the number of amino acid substitutions per site.

**Figure 5 insects-16-00356-f005:**
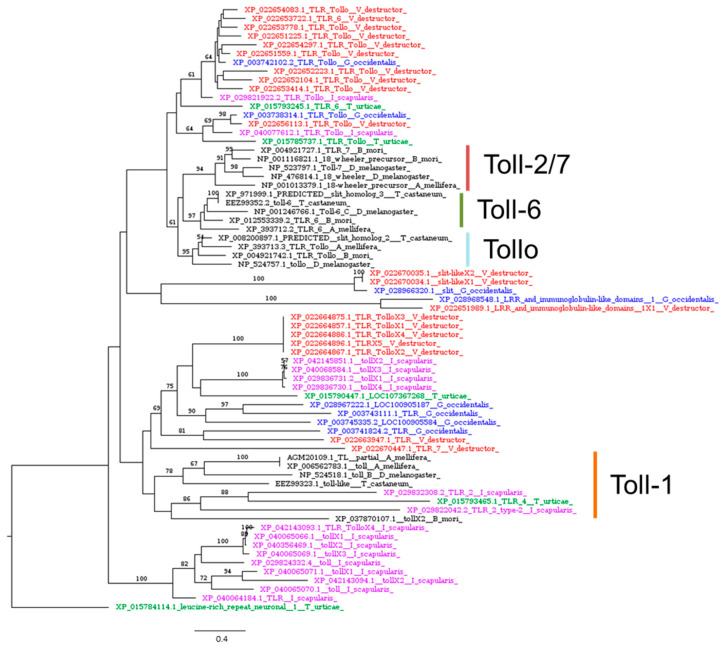
Phylogenetic tree based on maximum likelihood analysis of Toll-1, Toll-6, Toll-7, and Tollo putative homologs identified in the genomes of *V. destructor* (in red), *G. occidentalis* (in blue), *T. urticae* (in green), and *I. scapularis* (in purple). The longest branch of the unrooted tree is used as the outgroup. Bootstrap support values > 50% are indicated at each node. The scale bar indicates the number of amino acid substitutions per site.

**Figure 6 insects-16-00356-f006:**
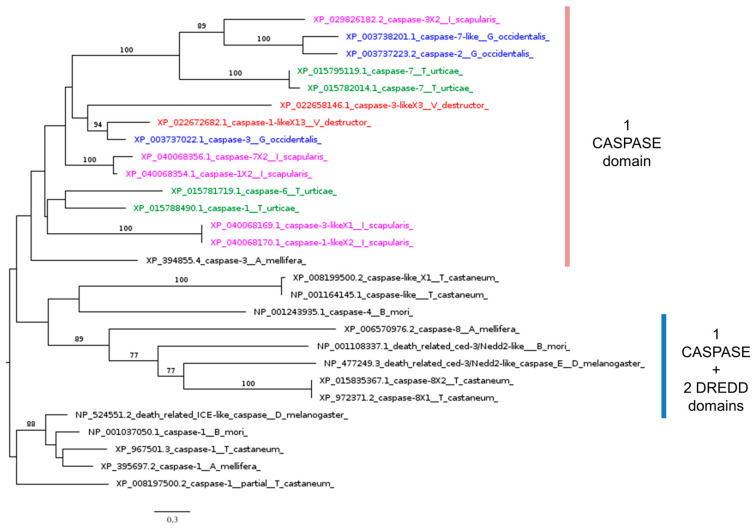
Phylogenetic tree based on maximum likelihood analysis of Dredd putative homologs identified in the genomes of *V. destructor* (in red), *G. occidentalis* (in blue), *T. urticae* (in green), and *I. scapularis* (in purple). The Drice homolog sequences are used as the outgroup. Bootstrap support values > 50% are indicated at each node. The scale bar indicates the number of amino acid substitutions per site.

**Figure 7 insects-16-00356-f007:**
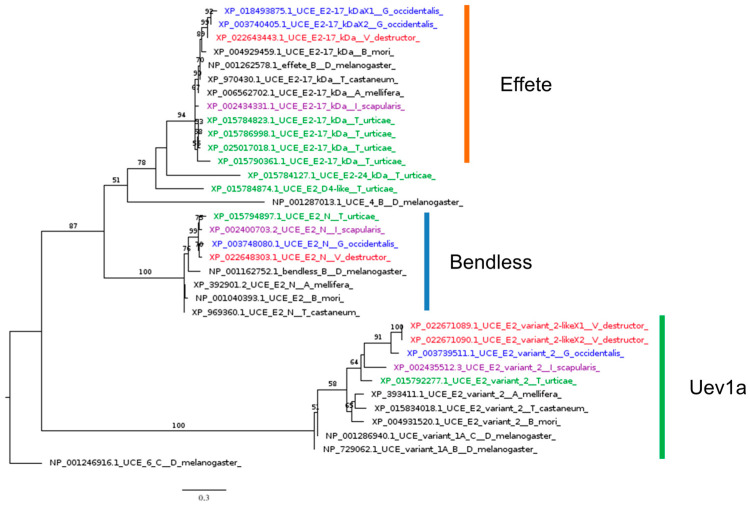
Phylogenetic tree based on maximum likelihood analysis of ubiquitination-conjugating enzymes (UCEs) Bendless, Effete, and Uev1a, identified in the genomes of *V. destructor* (in red), *G. occidentalis* (in blue), *T. urticae* (in green), and *I. scapularis* (in purple). The longest branch of the unrooted tree is used as the outgroup. Bootstrap support values > 50% are indicated at each node. The scale bar indicates the number of amino acid substitutions per site.

**Figure 8 insects-16-00356-f008:**
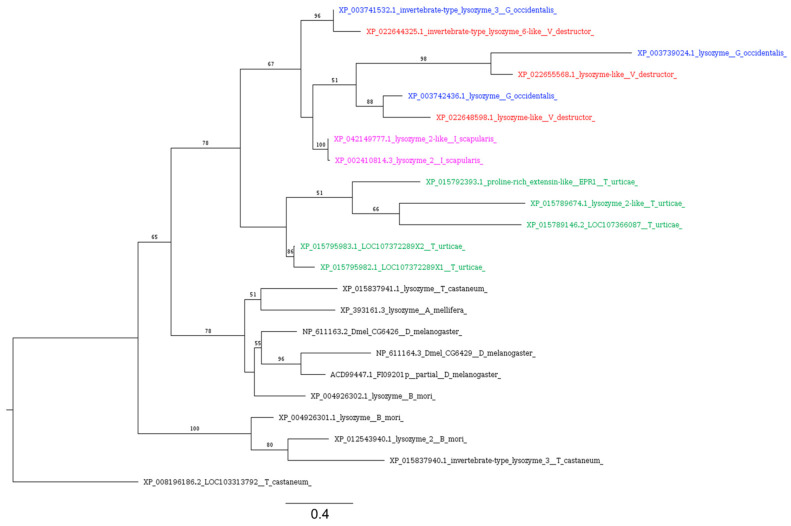
Phylogenetic tree based on maximum likelihood analysis of i-type lysozymes identified in the genomes of *V. destructor* (in red), G. occidentalis (in blue), *T. urticae* (in green), and *I. scapularis* (in purple). The longest branch of the unrooted tree is used as the outgroup. Bootstrap support values > 50% are indicated at each node. The scale bar indicates the number of amino acid substitutions per site.

## Data Availability

All FASTA sequences used, alignments, and tree files are available from Zenodo at https://doi.org/10.5281/zenodo.14837927.

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
