# Peer review of "Immune-Related Genes in the Honey Bee Mite Varroa destructor (Acarina, Parasitidae)"

_insects, 2025, doi:10.3390/insects16040356_

Round 1

Reviewer 1 Report

Comments and Suggestions for Authors

The current study characterized the immune gene set of Varroa destructor by inferring protein homology based on sequence similarity through BLASTp searches and compared it with related tick and mite species. These findings significantly improve the current knowledge of immune genes in this species and lay the foundation for functional studies aimed at clarifying immune responses in mites. Several genes have been investigated with specific comparisons among different Acarina species, considering the most updated and relevant references.

However, I still have some concerns listed below:

  • In the introduction (lines 58-78) RNAi was described as a selective and sustainable approach to control destructor and it is hoped that the results obtained in this work can be used in the future to develop RNAi assays against immune genes of this species. Has this approach been applied to other mite species? Are there any immune system genes that have proven effective as targets for RNAi-based control in other pest species? I would suggest discussing this point, considering the background outlined in the introduction.
  • Review the text (from line 141 to line 472) including captions and references, for the use of italics for species names.
  • Tables 1,2 and 3 are not easy to read. I suggest remaking them by eliminating some columns. Then the genes not found in destructor could be removed from the tables and listed separately.
  • Line 47-49: Varroa can also impact on reproductive fitness, homing and orientation abilities and immune response of bees, for example. I suggest to integrate this sentence and add references.
  • Line 75: Better say it did not affect mite survival.
  • Figure 1, line 135: I do not see any numbers in the figure. What do the empty, filled, and half-filled boxes represent?
  • Figure 2: it is not mentioned anywhere. Could it be placed in line 162?
  • Line 224: Better say five transcripts were identified in Drosophila.
  • Line 246: Figure 4.
  • Line 266: Better say it cannot be considered.
  • Line 281: Better say Palmer and Jiggins [26] failed to find a Tube homolog.
  • Reference 3, lines 516-518: it is incomplete (journal, pages..).
  • Reference 47, lines 624-626: it is incomplete (journal, pages..).
Comments on the Quality of English Language

The writing is not so clear and fluid. I recommend reviewing the English.

Author Response

Dear Reviewer 1,

here we have attached a point by point response (.doc).

We really appreciate  your suggestion which greately enhanced our manuscript.

Reviewer 2 Report

Comments and Suggestions for Authors

In this study, the authors investigated the immune gene repertoire of V. destructor by querying its NCBI nr protein database. They identified a large potential gene set for bee mite control. However, the authors' current results do not represent meaningful advances in this area.
Major:
    1.    Simply using blastp sequence comparison seems to be insufficient and provides too little information. Is consideration given to RT-PCR validation of potential candidate genes for mite suppression to clarify the spatial characterization of their expression and to provide a basis for ultimately obtaining the desired effect.
    2.    As far as I can see, the authors provide a large set of candidate mite-resistant genes, but in the course of their analysis do not give convincing enough evidence that these genes have a significant mite-suppressive effect. Logically, the fact that a gene plays a role in a life process is not the same as the fact that interfering with its expression is lethal, and the authors provide such a large number of potential candidate genes that, in a sense, such a study has limited significance.
Minor:
    3.    The readability of the table is average, and it is recommended that the genes that the authors consider to be candidates be placed more forward, or even a different way of visualizing the data.

Author Response

Dear Reviewer 2,

here we have attached a point by point response (.doc).

We really appreciate  your suggestion which greately enhanced our manuscript.

Reviewer 3 Report

Comments and Suggestions for Authors

This ms dealt with identifying the immune genes in bee mite Vorroa destructor, which lay basis for mite control in future.
but there are still some points needed to consider before its acceptance for publication in insects. 

abstracts:
1. authors give little description of immune genes that mite has, but too much information of immune genes that mite lack or does not have. 

2. while in results, you find many immune genes in the bee mite, do you validate the genes in the genome via PCR or sequencing? Obviously, the authors only did the in silico bioinformation analysis. I would rather to see more data for validation of these immune genes present in the bee mite genome. 

3 and another tough question to think about. How many mite genomes were sequenced and were used for your analysis. The immune genes were not found due to limited genome resource or limited depth of sequencing? How to validate these immune genes absent or lacked.

results
figure1, more detail should be given as to the empty/filled block? no size variation for triangles=they lack same number of immune genes across groups?

tables : text front size are too big that make the tables unfit to the page, it is necessary to make the table compacted.

no discussion part?

Author Response

Dear Reviewer 3,

here we have attached a point by point response (.doc).

We really appreciate  your suggestion which greately enhanced our manuscript.
